# Geometry-Based Computational Fluid Dynamic Model for Predicting the Biological Behavior of Bone Tissue Engineering Scaffolds

**DOI:** 10.3390/jfb13030104

**Published:** 2022-07-27

**Authors:** Abdalla M. Omar, Mohamed H. Hassan, Evangelos Daskalakis, Gokhan Ates, Charlie J. Bright, Zhanyan Xu, Emily J. Powell, Wajira Mirihanage, Paulo J. D. S. Bartolo

**Affiliations:** 1Department of Mechanical, Aerospace and Civil Engineering, University of Manchester, Manchester M13 9PL, UK; mohamed.hassan@manchester.ac.uk (M.H.H.); evangelos.daskalakis@manchester.ac.uk (E.D.); gokhan.ates@manchester.ac.uk (G.A.); charlie.bright@manchester.ac.uk (C.J.B.); zhanyan.xu@manchester.ac.uk (Z.X.); emily.powell-2@manchester.ac.uk (E.J.P.); 2Department of Materials, The University of Manchester, Manchester M13 9PL, UK; wajira.mirihanage@manchester.ac.uk; 3Singapore Centre for 3D Printing, School of Mechanical and Aerospace Engineering, Nanyang Technological University, Singapore 639798, Singapore

**Keywords:** additive manufacturing, bone scaffolds, cell viability, computational fluid dynamics, scaffold geometry

## Abstract

The use of biocompatible and biodegradable porous scaffolds produced via additive manufacturing is one of the most common approaches in tissue engineering. The geometric design of tissue engineering scaffolds (e.g., pore size, pore shape, and pore distribution) has a significant impact on their biological behavior. Fluid flow dynamics are important for understanding blood flow through a porous structure, as they determine the transport of nutrients and oxygen to cells and the flushing of toxic waste. The aim of this study is to investigate the impact of the scaffold architecture, pore size and distribution on its biological performance using Computational Fluid Dynamics (CFD). Different blood flow velocities (BFV) induce wall shear stresses (WSS) on cells. WSS values above 30 mPa are detrimental to their growth. In this study, two scaffold designs were considered: rectangular scaffolds with uniform square pores (300, 350, and 450 µm), and anatomically designed circular scaffolds with a bone-like structure and pore size gradient (476–979 µm). The anatomically designed scaffolds provided the best fluid flow conditions, suggesting a 24.21% improvement in the biological performance compared to the rectangular scaffolds. The numerical observations are aligned with those of previously reported biological studies.

## 1. Introduction

Modern tissue engineering strategies rely on the use of additive manufacturing with biocompatible and biodegradable materials, living cells and growth factors to create constructs to restore, repair, and improve the function of damaged tissues and/or organs [1,2,3]. There are two commonly used strategies, cell-laden and scaffold-based approaches [4,5,6]. The cell-laden approach is based on the use of additive manufacturing to selectively print bio-inks (hydrogels containing cells) [7,8]. A range of hydrogels has been investigated, and the fabrication of multi-material and cellular constructs has been reported [9,10]. However, cell-laden constructs have limited mechanical properties, making them more suitable for soft-tissue applications [11,12]. Scaffolds are the most common approach used in bone tissue engineering [4,5].

Scaffolds are 3D biocompatible and biodegradable porous structures that provide the necessary substrate for cell attachment, proliferation, differentiation, extracellular matrix (ECM) formation, and mineralisation [6,7]. They must be highly porous structures with fully interconnected pores to allow cell growth, and biodegradable with degradation kinetics similar to the tissue regeneration rate [4,5,6,7]. Interconnected pores form channels which allow for the transfer of nutrients and waste and encourage the formation of ECM by enhancing cellular functions [8,9]. Moreover, scaffolds must be designed to have mechanical properties similar to those of native tissue, allowing them to withstand loads during the regeneration process [10,11].

The performance of a scaffold depends on the printing material, parameters, and topological characteristics. A range of polymeric (e.g., polycaprolactone, polylactic acid, poly glycolic acid), ceramic (e.g., hydroxyapatite, tricalcium phosphate), and composite materials have been investigated and their selection depends on the mechanical, degradability and biological properties [12,13]. Processing parameters determine the morphological development during the printing processes, influencing crystallinity, crystal orientation, and crystal size, which affect the biomechanical properties of the scaffold [14,15,16]. Moreover, pore size, pore shape, pore distribution, and filament diameter determine both the mechanical and biological properties of the scaffold, and how cells interact with it [17,18,19,20]. Once printed, scaffolds are seeded with cells and pre-cultured in a bioreactor before implantation or directly implanted into the defect area. In both cases, scaffold permeability is a critical parameter.

Design and fabrication parameters are chosen considering blood transport through the scaffold. Blood flow in the scaffold is essential to deliver the necessary oxygen and nutrients to the cells and to remove deoxygenated blood and waste [21,22]. Biological behavior is mainly dependent on the blood-scaffold interaction, resulting in wall shear stress (WSS) which is an important stimulus for cells. Experimental investigation of blood flow through scaffolds has many constraints including time, cost and obtaining accurate results without flow disruption. These constraints can be resolved using computational simulations.

A common solution is the use of Finite Element Analysis (FEA) and CFD is a popular approach in the field of bioengineering to predict the mechano-biological properties of scaffolds [23,24,25]. CFD is essential to understand the in vivo performance of a scaffold and to investigate its permeability characteristics, which strongly determine its biological performance [19,20,21,22]. Haemodynamic metrics, such as WSS and fluid pressure, are known to impact the nitric oxide levels within the scaffold which is associated with the mechanical stimulus on cells [23,24]. WSS contributes to the differentiation of pluripotent stem cells into endothelial, cardiac, haematopoietic, and osteoblast phenotypes [25,26,27,28,29]. Reports suggest that WSS in the range of 0–30 mPa stimulate the overall biological activity of mesenchymal stromal cells (MSCs), in the range of 0.11–10 mPa stimulates osteogenic differentiation, and in the range of 0.55–24 mPa stimulates the mineralisation process of bone cells [26,27,28,29,30]. However, WSS values above 60 mPa are linked to cell death [22,31,32,33,34,35,36].

This study proposes a simple and inexpensive simulation strategy to investigate the biological and haemodynamic performance of 3D printed bone scaffolds with different configurations and pore sizes, considering the following:Use of different blood flow velocities (BFV) through scaffolds that are used for large-sized defects located in bones [37,38,39].Newtonian fluid: fluid is assumed to have a constant viscosity to simplify the system. It is noted that WSS value distribution can change when using a non-Newtonian fluid [40,41].Surface roughness: The surface was assumed to be smooth based on previous experimental data, as it is dependent on the fabrication method. It also must be noted that including roughness can change how the fluid interacts with the scaffolds [42].

## 2. Modelling and Simulation

### 2.1. Scaffold Design

Two scaffold configurations were considered for simulations purposes based on previously published experimental work for two scaffold design configurations [43,44]. The first configuration (Case 1) corresponds to rectangular scaffolds designed with a 0/900 lay-down pattern (Figure 1) and three different pore sizes (Case 1A:300 µm pore size; Case 1B:350 µm pore size; Case 1C:450 µm pore size). The scaffolds were designed with 12 filament layers, 350 µm in diameter, and 330 µm in layer thickness. The second configuration (Case 2) corresponded to anatomically designed scaffolds with six rings (Figure 2) and a graded pore size (first ring, 476 µm; second ring, 629 µm; third ring, 670 µm; fourth ring, 730 µm; fifth ring, 803 µm; sixth ring, 979 µm). The scaffold porosity was determined using the following equation:(1)$$Porosity=\frac{V_{Scaffold}}{V_{Solid}}$$
where *V_scaffold_* is the volume of the scaffold filaments, and *V_solid_* is the volume of the solid part. The specific surface area (*S*), which defines the available area for cell attachment, proliferation, and differentiation and influences permeability (increases with increasing pore size and decreasing *S*) was calculated as follows:(2)$$s=\frac{SA}{V}$$
where *SA* is the surface area and *V* is the volume. The geometric characteristics of the scaffolds are listed in Table 1.

### 2.2. Simulation

CFD simulations were conducted to investigate the fluid flow characteristics of different scaffolds, including permeability, WSS, and flow velocity. In contrast to other CFD studies that have considered water as the fluid, this study used blood as the fluid medium (Table 2). The fluid was considered incompressible and Newtonian, and the scaffolds were modelled as rigid bodies within the fluid domain [36,37,38]. Moreover, the 3D Navier–Stokes equations were considered to solve the conservation of mass and momentum across the scaffold using the finite volume method. The steady state Navier–Stokes equations are given by [45]:(3)$$\rho \frac{\partial u}{\partial t}-\mu \nabla^{2}u+\rho \left(u\cdot \;\nabla \right)u+\nabla p=F$$
(4)∇·u=0 
where *ρ* represents the fluid density (kg/m^3^), *t* is the time (s), *u* is the fluid velocity (m/s), *µ* is the dynamic viscosity (Pa.s), ∇ represents the del operator, *p* is the pressure (Pa) and *F* denotes other forces (i.e., gravity or centrifugal force) acting in the fluid domain. In this study, *F* is assumed to be zero [46,47].

The intrinsic permeability of the scaffolds, *k*_0_, (m^2^), for a range of inlet velocities is given by [48]:(5)$$k_{0}=\frac{u\mu L}{\Delta P}\;$$
where *L* is the scaffold length (m) and Δ*P* is the pressure drop (Pa). However, it should be noted that for very high inlet velocities, Darcy’s law is inadequate for estimating the permeability of the scaffold [24,39,40,41]. Typically, when the calculated interstitial Reynolds number (*Re*) is higher than 8.6, Darcy’s classic equation (Equation (5)) is invalid [39,42]. Therefore, to evaluate the validity of Darcy’s law, the Reynolds number (*Re*) was determined as follows [28,41]:(6)$$Re=\frac{\rho ud}{\mu }\;$$
where *d* is the pore size (m).

Assuming a laminar flow regime, the WSS can be calculated throughout the domain for each scaffold by considering the normal velocity gradient over a filament [49,50]:(7)$$\tau_{w}=\mu \frac{\partial u}{\partial n}\;$$
where *τω* represents the wall shear stress (Pa) and *n* indicates the x-, y-, and z-directions (m).

CFD simulations were conducted using the finite volume method (FVM) using the ANSYS CFX solver (ANSYS INC, Washington, PA, USA). The scaffolds were designed using SolidWorks (Dassault Systems, Vélizy-Villacoublay, France) and the CAD models were imported into the ANSYS CFX design modeller, where the filaments were fused to create a single body. A cylindrical enclosure was created around the scaffold to simulate a flow in a bone line environment. The fluid domain was constructed via a Boolean subtraction operation using the scaffold as a tool and a cylinder with a radius of 0.001 m and 0.01 m on either side of the centre of the scaffold.

The inlet was defined at one end of the cylindrical domain, with an outlet on the opposite side. The external region of the flow domain is defined as a rigid-bodied wall (no-slip condition). The domain was meshed using CFX’s built-in mesher using the patch-conforming method with tetrahedral elements. A mesh independence study was conducted for the three domain refinements, resulting in differences of less than 5%. The total number of elements for the considered scaffolds was 3,441,553 for Case 1A, 4,180,282 for Case 1B, 4,154,355 for Case 1C, and 6,384,572 for Case 2.

To mimic physiological conditions, both isothermal heat transfer at 37 °C and a shear stress transport (SST) turbulence model were considered. The SST model is a two-equation k-w model that improves the prediction of separation in the near-wall regions of the model [45]. Therefore, the model can account for turbulence which is an physiological characteristic of blood flow [51].

In the pre-processor the inlet was defined with medium (5%) turbulence, and a range of inlet blood flow velocities (1–9 mm/s) were used following experimental conditions simulating tibia bone fractures [52]. The outlet was set as a 0-resistance boundary condition. The convergence criteria for the residuals were set to a root-mean-square value of 1e-4 using second order methods. Each simulation was completed on an Intel-Xeon quad-core CPU using the MPI. The results were post-processed using the CFX CFD-post program to extract the velocity, pressure, and WSS values. A histogram of the WSS versus the geometrical percentage was also exported.

## 3. Results and Discussion

Figure 3 shows the pressure drop variation as a function of inlet velocity for the different scaffolds. Results were obtained by considering two virtual probes on the top and bottom of the scaffolds and finding the corresponding value from a plane contour passing through the scaffold. As observed, the highest-pressure drop was obtained for the Case 1A and Case 1C scaffolds, followed by Cases 1B and 2. These results can be explained by the combined effect of pore size and architecture of the scaffold (number of filaments).

The high pressure drop observed for Case 1A can be explained by the larger interaction between the fluid and the scaffold due to the large number of filaments, high specific area, and the smaller diameter, while for Case 1C the results are due to the low interactions (among the rectangular scaffolds this is the configuration presenting the lowest number of filaments and the lowest specific surface area) and large pore size. The gradient architecture in Case 2 explains the lowest pressure drop.

Using the pressure drop values, it was possible to estimate permeability using Darcy’s law, and the results showed that Case 2 scaffolds presented the highest permeability, followed by Cases 1 B, 1A, and 1C (Figure 4). Moreover, the different in permeability decreases with the increase in blood flow velocity, indicating a larger effect of pore size and geometries at lower velocity compared to higher velocity. Nevertheless, the trend is consistent with different velocities applied. As observed, the anatomically designed scaffolds (Case 2) guaranteed sufficient blood flow through the scaffolds, whereas among the rectangular scaffolds, the best results were achieved with Case 1B.

Figure 5, Figure 6, Figure 7, Figure 8, Figure 9, Figure 10, Figure 11, Figure 12, Figure 13, Figure 14, Figure 15 and Figure 16 show the changes in the fluid metrics (velocity, pressure, and WSS) as a function of the inlet velocity. As observed, for rectangular scaffolds, the pressure increases with increasing pore size (Case 1C presents the highest-pressure values followed by Cases 1A and Case 1B) (Figure 5, Figure 6 and Figure 7). However, in the case of scaffolds with a gradient of pore sizes (Case 2), regions with higher average pore sizes presented lower pressure values (Figure 8). For all considered scaffolds, results show that an increase in the inlet velocity results in larger values of pressure due to the larger deflection forces occurring when the fluid encounters the scaffolds of higher pore sizes and lower specific surface areas. For the velocity, the results suggest no impact of pore size (Figure 9, Figure 10, Figure 11 and Figure 12).

For the WSS (Figure 13, Figure 14, Figure 15 and Figure 16), the maximum value was observed in Case 1B, followed by Cases 1C, 1A, and 2. In Cases 1A and 1B, high WSS values were observed at the top of the scaffold, with a significant drop from the top surface to the inner zone. However, in Case 1C, high WSS values were observed on both the top and side faces of the scaffold. This can be attributed to the scaffold geometry and specific area, suggesting a correlation between a higher WSS and lower specific surface area. As WSS has a major impact on the biological performance of scaffolds, the results suggest that the specific surface area and pore size are key design parameters. Moreover, the maximum pressure was incident on the outer edges of all scaffolds; however, it was relatively lower for Case 2, which explains the overall lower WSS values observed on the anatomically designed scaffolds (Figure 16). Therefore, to reduce the intensity of pressure on the sides of the scaffolds, Case 2 scaffolds can be used because they present circular geometries and a higher specific area, resulting in a reduction in the pressure at the top surface and overall pressure. Thus, the results suggest that the Case 2 scaffolds present an improved overall biological behavior compared to the more commonly used rectangular scaffolds (Case 1).

To further analyse the WSS distribution, the percentage of the total surface area of the scaffolds with WSS values lower than 30 mPa against the inlet velocity was also considered (Figure 17). This WSS value was used as a reference, as it is the cut-off value for cell proliferation and differentiation. As observed for all scaffolds, for velocities ranging between 1 and 5 mm/s the occurrence of WSS > 30 mPa was very low, between 5 and 7 mm/s the occurrence of WSS > 30 mPa slightly increased, and for velocities ranging between 7 and 9 mm/s the occurrence of WSS > 30 mPa significantly increased. The highest WSS values were observed for Case 1C, followed by Cases 1A, Case 1B and Case 2. The average WSS values are listed in Table 3. In the case of rectangular scaffolds, the results suggest that the highest average cell viability is expected to occur in Case 1B scaffolds (the scaffold percentage of volume with WSS < 30 mPa is 61.64%), followed by Case 1C (61.56%), and Case 1A (59.44%). Among all scaffold architectures, the highest cell viability was expected to occur in Case 2 (85.86%), with the highest WSS values at the sides and the lowest WSS at the centre.

These results agree with experimental studies that investigated the biological behavior of similar rectangular scaffolds, showing that the Case 1B scaffolds presented the best results [43,44]. Other experimental studies also agree with the improved biological behavior that anatomically designed scaffolds (Case 2) present compared to rectangular scaffolds (Case 1) [43,44].

The strategy of this study differs from other published works in that it uses a simple design by controlling the pore size and geometry by changing the fibre size and distance. Other studies have used a unit cell approach that may have an irregular geometrical flow, as they are based on metallic AM. However, our findings agree with the literature using similar assumptions that permeability and WSS are geometry-dependent, and that there is no clear trend with pore sizes and geometries.

This strategy is based on previously published experimental studies using extrusion-based additive manufacturing with polymers and polymer composites. The use of anatomically designed scaffolds (Case 2) is a new and promising strategy to replace conventional rectangular scaffolds. Previously published experimental and numerical results obtained in this study can be used to optimize different scaffold geometries, reducing the time and cost of repeated biological experiments. Future perspectives may include the use of graded porous scaffolds and non-Newtonian and transient simulations to further develop numerical strategies for bone tissue engineering scaffolds.

## 4. Conclusions

Different scaffold geometries have been investigated using Computational Fluid Dynamics (CFD) to understand the effect of pore size and geometry on biological behavior which is usually difficult to obtain experimentally. Among the rectangular scaffolds, Case 1B exhibited the optimum balance between pore size and specific surface area, showing the highest permeability, WSS < 30 mPa at higher velocities, and a lower pressure drop. Anatomically designed scaffolds (Case 2) presented similar velocities but improved WSS distribution, suggesting superior biological behavior. The numerical results were aligned with previously obtained experimental results.

In the future, a simulation model will be developed using 3D scanning data of printed scaffolds to better describe the architecture of the scaffolds, including surface roughness which may impact the results. Additionally, transient simulations, advanced fluid-structure interaction models, and advanced fluid rheology models, including the use of pseudoplastic models for the blood model, were considered.

## Figures and Tables

**Figure 1 jfb-13-00104-f001:**
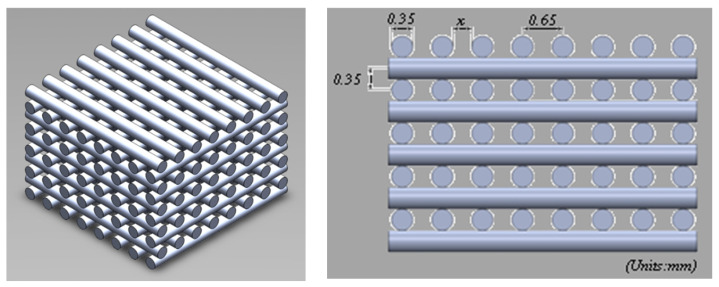
Case 1: Geometric representation of rectangular scaffolds.

**Figure 2 jfb-13-00104-f002:**
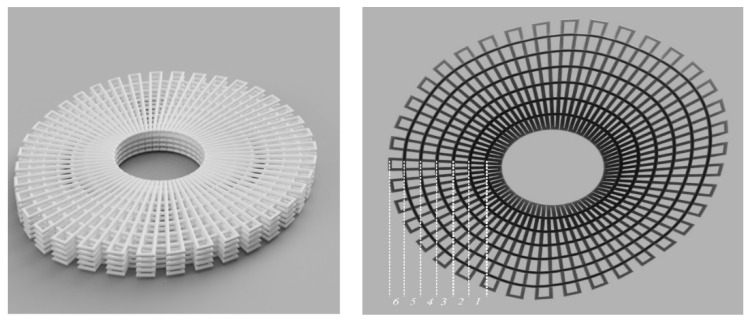
Case 2: Geometric representation of anatomically designed scaffolds, and the numbers show the rings in the design.

**Figure 3 jfb-13-00104-f003:**
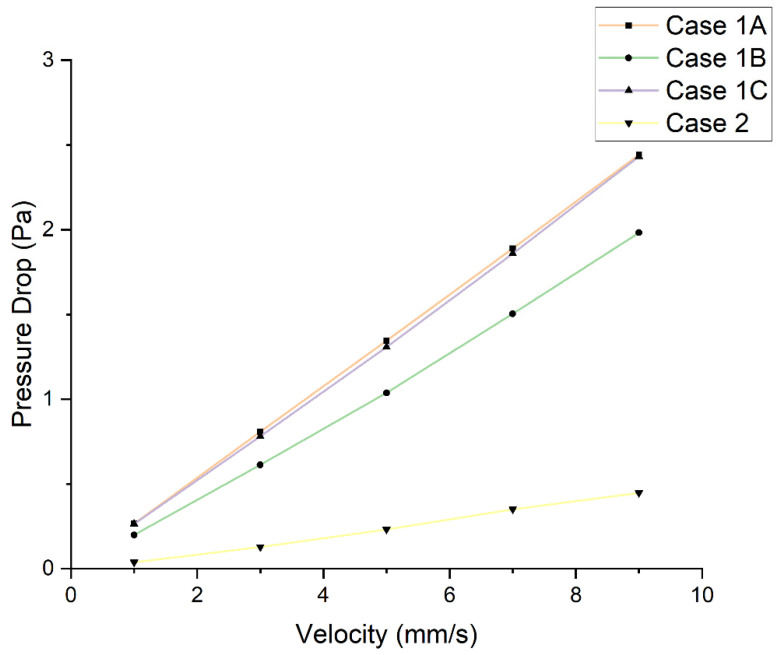
Effect of different inlet velocities on the pressure drop for different scaffolds.

**Figure 4 jfb-13-00104-f004:**
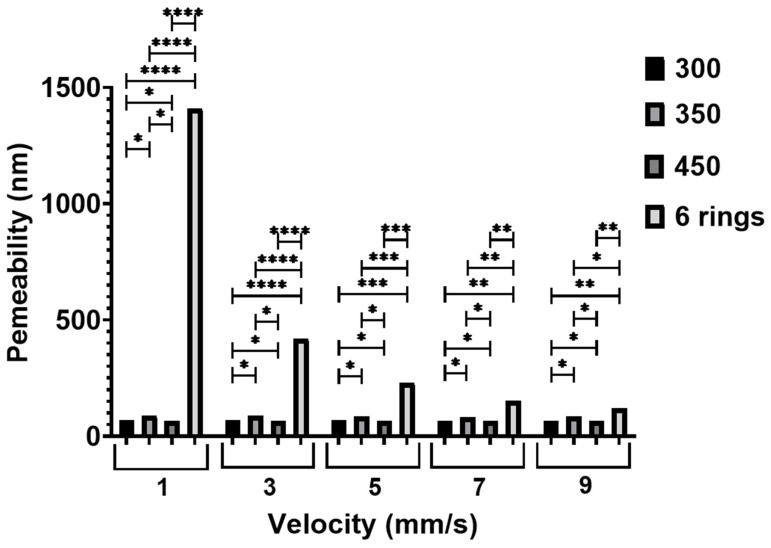
The effect of inlet velocity on the permeability. * Statistical evidence (*p* < 0.05) analysed by one-way ANOVA, and Tukey post hoc test. The * statistical evidence (*p* < 0.05), **, *** and **** is the one-way analysis of variance (one-way ANOVA) and Tukey’s post hoc test with the use of GraphPad Prism software and is used to show the difference between the results. The * is a small difference, while more * are added as the differences between the results increases.

**Figure 5 jfb-13-00104-f005:**
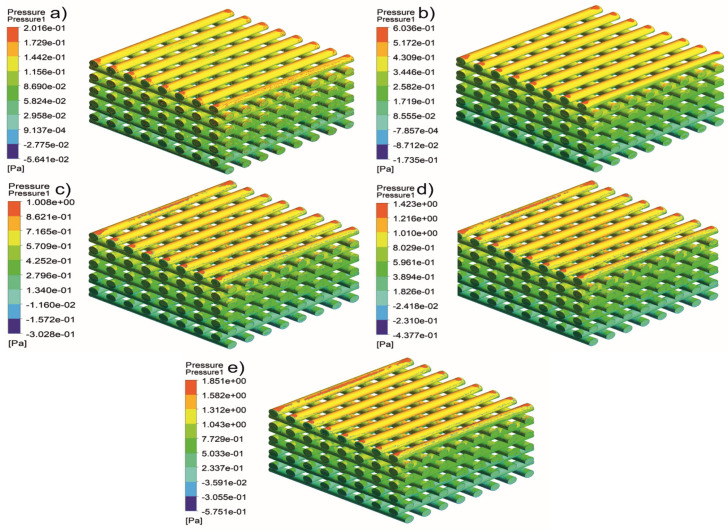
Case 1A: pressure contours for different inlet velocities. (**a**) 1 mm/s, (**b**) 3 mm/s, (**c**) 5 mm/s, (**d**) 7 mm/s, and (**e**) 9 mm/s.

**Figure 6 jfb-13-00104-f006:**
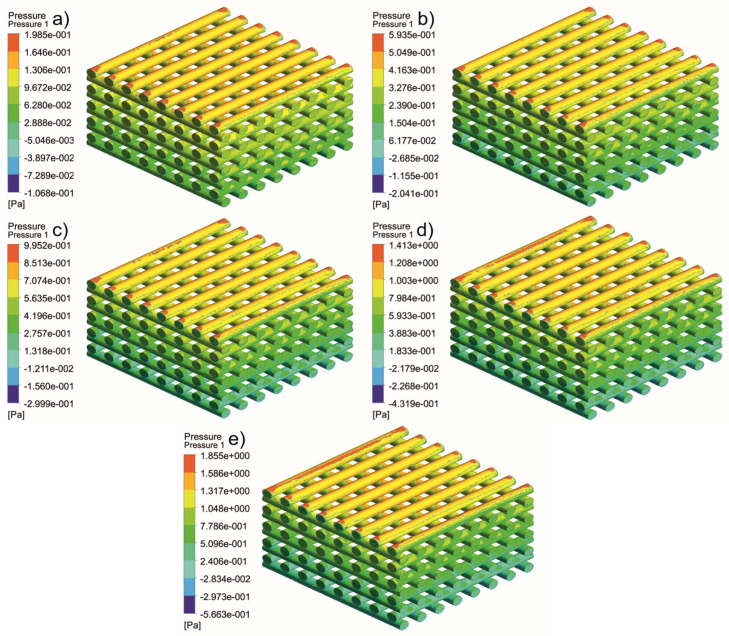
Case 1B: pressure contours for different inlet velocities. (**a**) 1 mm/s, (**b**) 3 mm/s, (**c**) 5 mm/s, (**d**) 7 mm/s, and (**e**) 9 mm/s.

**Figure 7 jfb-13-00104-f007:**
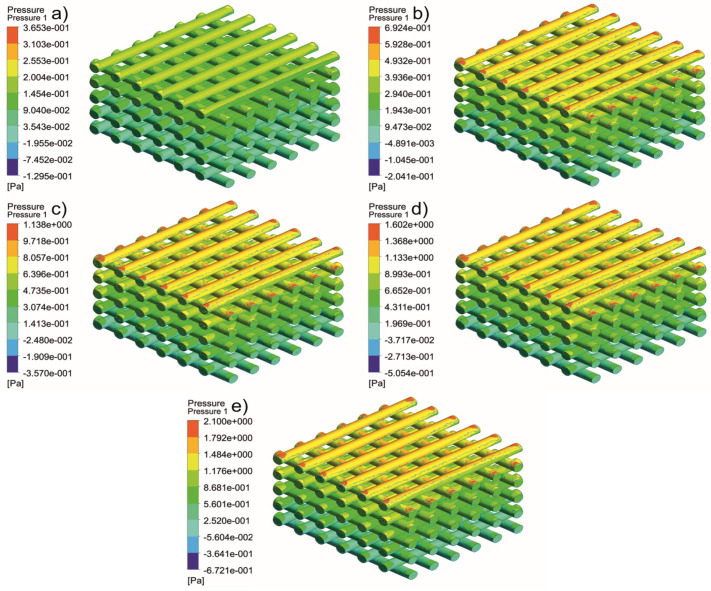
Case 1C: pressure contours for different inlet velocities. (**a**) 1 mm/s, (**b**) 3 mm/s, (**c**) 5 mm/s, (**d**) 7 mm/s, and (**e**) 9 mm/s.

**Figure 8 jfb-13-00104-f008:**
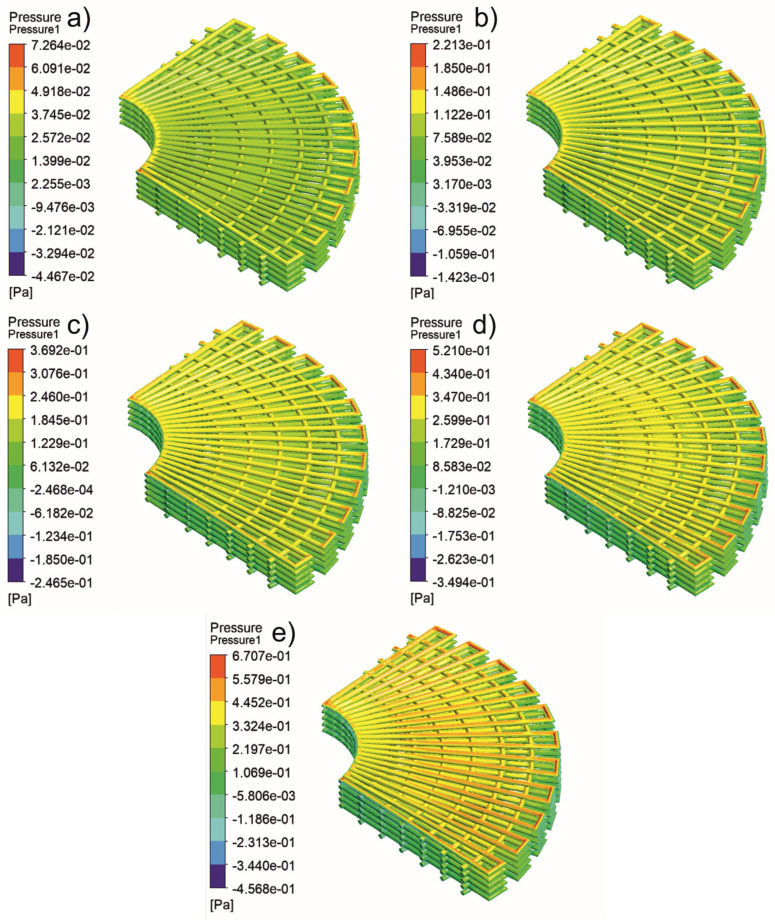
Case 2: pressure contours for different inlet velocities. (**a**) 1 mm/s, (**b**) 3 mm/s, (**c**) 5 mm/s, (**d**) 7 mm/s, and (**e**) 9 mm/s.

**Figure 9 jfb-13-00104-f009:**
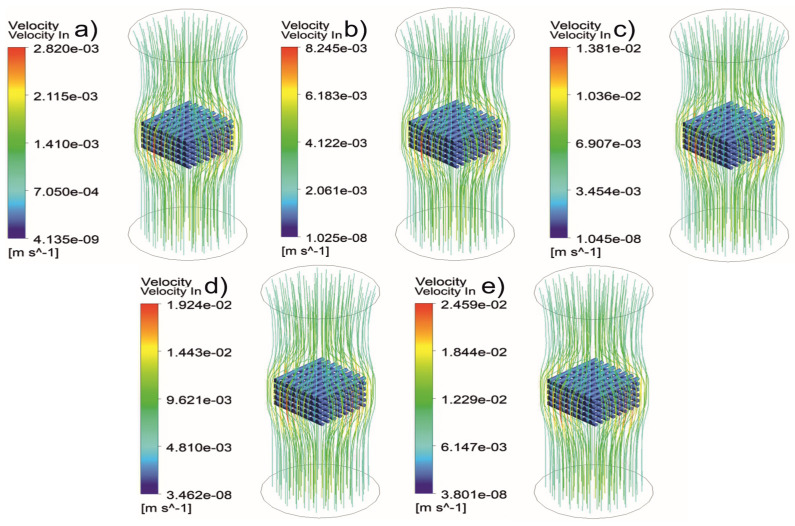
Case 1A: velocity streamlines for different inlet velocities. (**a**) 1 mm/s, (**b**) 3 mm/s, (**c**) 5 mm/s, (**d**) 7 mm/s, and (**e**) 9 mm/s.

**Figure 10 jfb-13-00104-f010:**
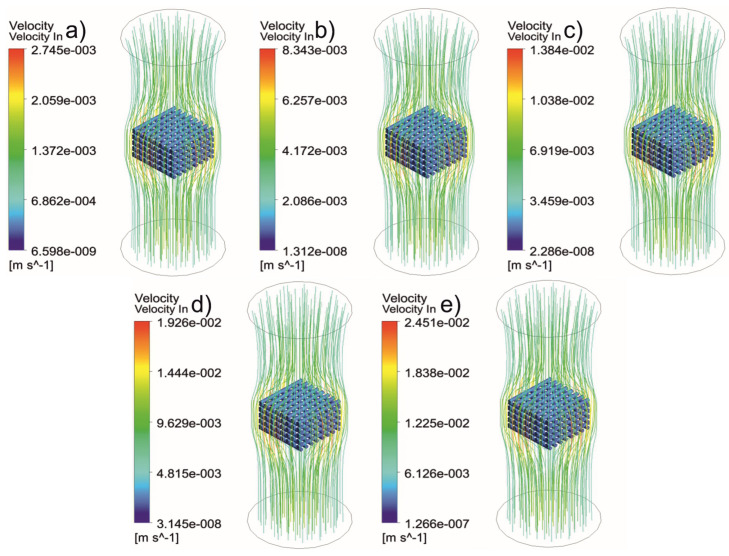
Case 1B: velocity streamlines for different inlet velocities. (**a**) 1 mm/s, (**b**) 3 mm/s, (**c**) 5 mm/s, (**d**) 7 mm/s, and (**e**) 9 mm/s.

**Figure 11 jfb-13-00104-f011:**
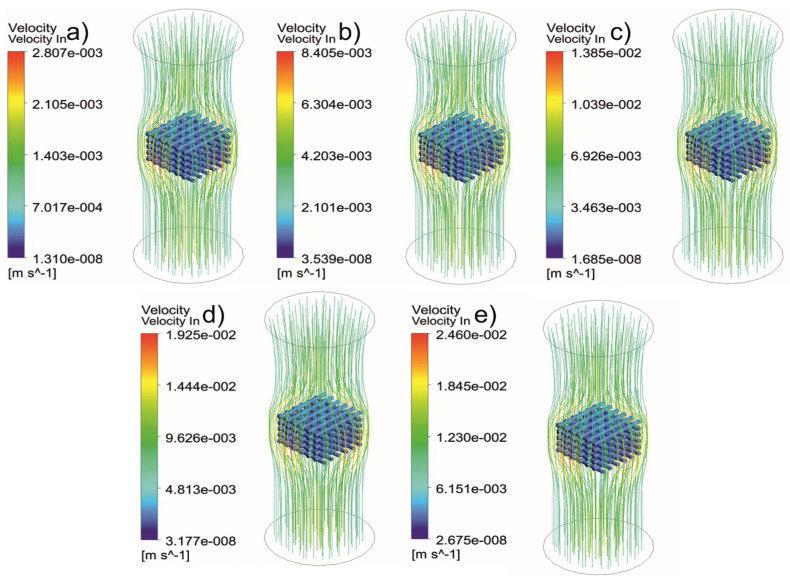
Case 1C: velocity streamlines for different inlet velocities. (**a**) 1 mm/s, (**b**) 3 mm/s, (**c**) 5 mm/s, (**d**) 7 mm/s, and (**e**) 9 mm/s.

**Figure 12 jfb-13-00104-f012:**
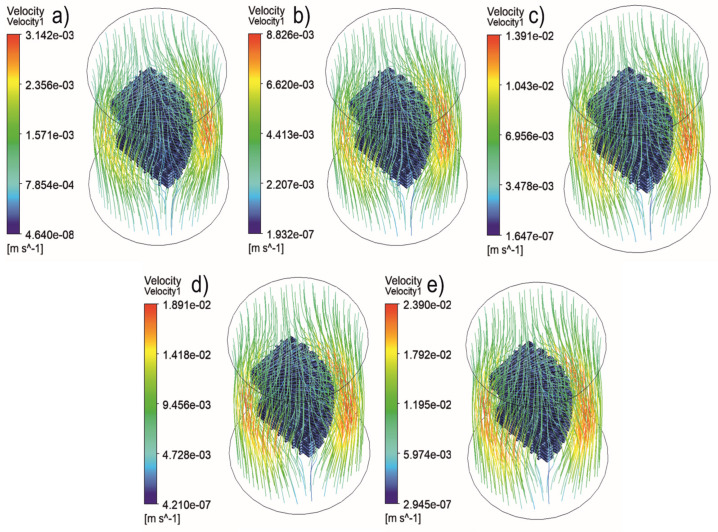
Case 2: velocity streamlines for different inlet velocities. (**a**) 1 mm/s, (**b**) 3 mm/s, (**c**) 5 mm/s, (**d**) 7 mm/s, and (**e**) 9 mm/s.

**Figure 13 jfb-13-00104-f013:**
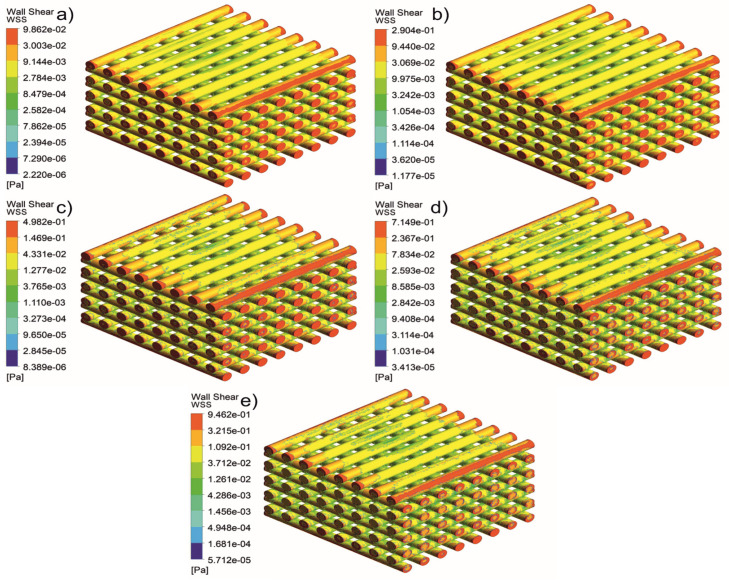
Case 1A: WSS for different inlet velocities. (**a**) 1 mm/s, (**b**) 3 mm/s, (**c**) 5 mm/s, (**d**) 7 mm/s, and (**e**) 9 mm/s.

**Figure 14 jfb-13-00104-f014:**
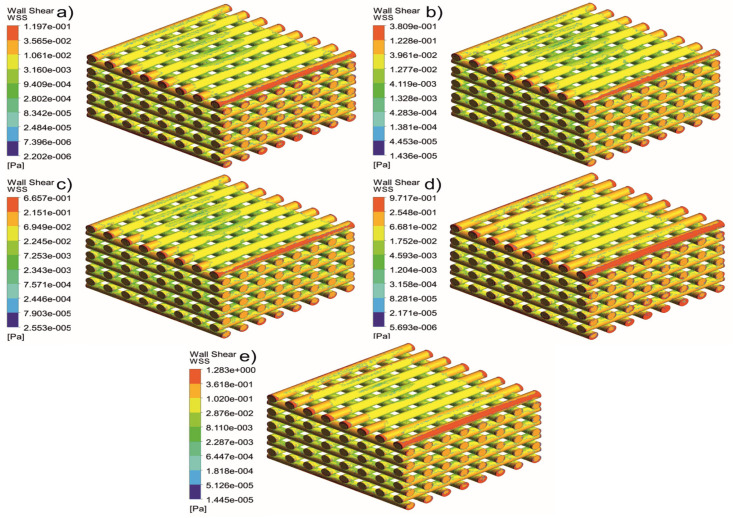
Case 1B: WSS for different inlet velocities. (**a**) 1 mm/s, (**b**) 3 mm/s, (**c**) 5 mm/s, (**d**) 7 mm/s, and (**e**) 9 mm/s.

**Figure 15 jfb-13-00104-f015:**
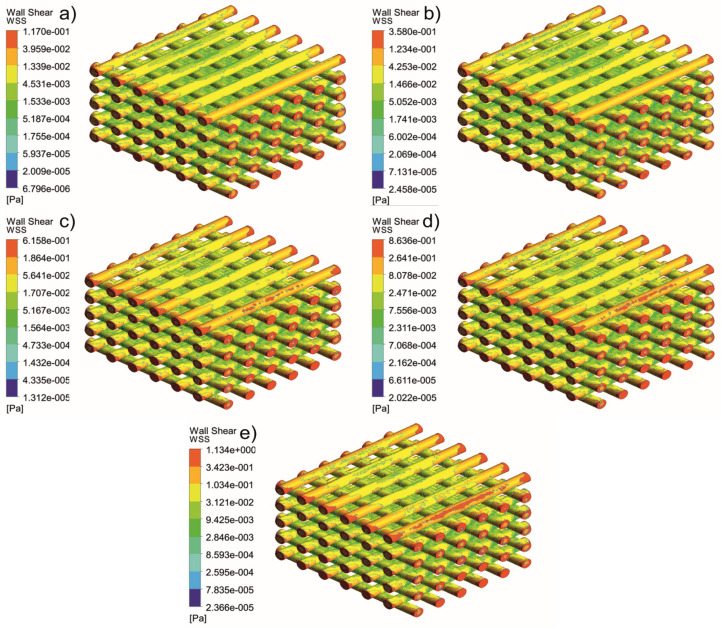
Case 1C: WSS for different inlet velocities. (**a**) 1 mm/s, (**b**) 3 mm/s, (**c**) 5 mm/s, (**d**) 7 mm/s, and (**e**) 9 mm/s.

**Figure 16 jfb-13-00104-f016:**
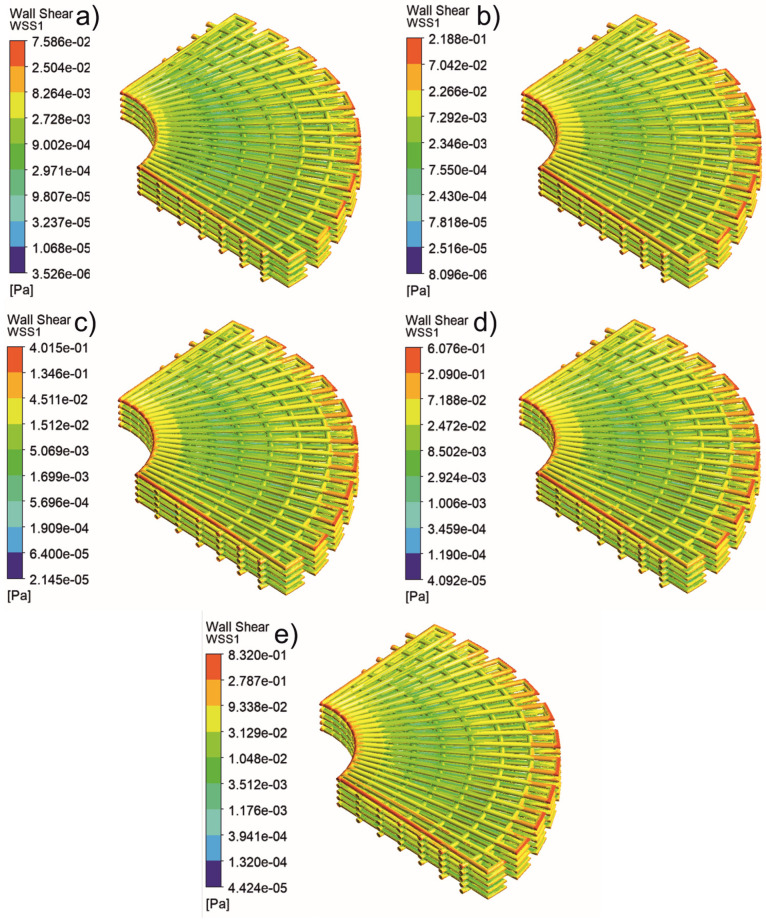
Case 2: WSS for different inlet velocities. (**a**) 1 mm/s, (**b**) 3 mm/s, (**c**) 5 mm/s, (**d**) 7 mm/s, and (**e**) 9 mm/s.

**Figure 17 jfb-13-00104-f017:**
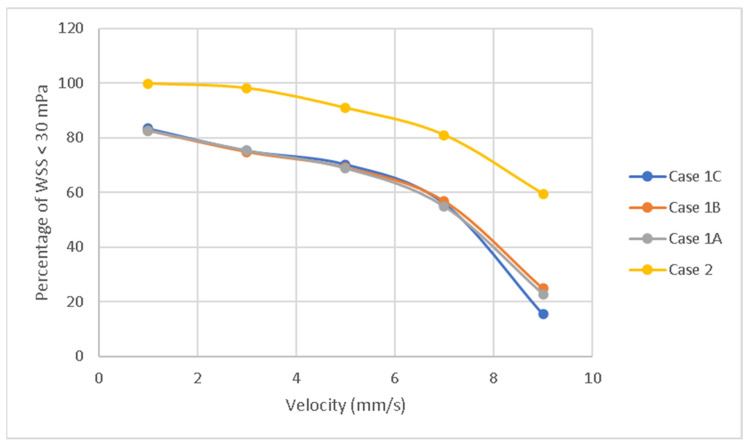
Cell viability indication: Percentage of volume presenting WSS values lower than 30 mPa for different inlet velocities and scaffold architectures.

**Table 1 jfb-13-00104-t001:** Geometrical characteristics of the scaffolds.

	Parameters	Value
**Case 1**	dimensions (mm)	20 × 20 × 3.1
	Number of layers	10
	Fibre diameter (mm)	0.33
	Pore size (mm)	0.300/0.350/0.45
	Porosity (%)	56.59/58.57/61.19
	Specific surface area (mm^−1^)	2.39/2.05/1.59
**Case 2**	Dimensions (mm)	31 × 26.7 × 3.1
	Number of layers	10
	Fibre diameter (mm)	0.33
	Pore size (mm)	0.476/0.629/0.670/0.730/0.803/0.979
	Porosity (%)	76.49
	Specific surface area (mm^−1^)	11.43

**Table 2 jfb-13-00104-t002:** Haemodynamic values used to simulate blood flow [45,46].

Parameters	Value
Molar Mass (kg kmol^−1^)	65,000
Density (kg m^−3^)	1056
Dynamic Viscosity (Pa.s)	0.0045
Blood Flow Velocity (BFV) (mm/s)	1, 3, 5, 7, 9
Heat capacity (kg m^−3^)/(J kg^−1^K^−1^)	1056/4000

**Table 3 jfb-13-00104-t003:** Average percentage of volume presenting WSS values lower than 30 mPa.

Velocity (mm/s)	Case 1C	Case 1B	Case 1A	Case 2
1	83.4	82.6	82.7	99.8
3	75.3	74.8	75.3	98.1
5	70.3	69.1	68.9	90.9
7	56	56.8	54.9	81
9	22.8	24.9	15.4	59.5
Average	61.56	61.64	59.44	85.86

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
