# Peer review of "Geometry-Based Computational Fluid Dynamic Model for Predicting the Biological Behavior of Bone Tissue Engineering Scaffolds"

_jfb, 2022, doi:10.3390/jfb13030104_

Round 1
Reviewer 1 Report
The work is concerned on CFD flow modeling in which the blood flows through some cases of the scaffold. In my opinion, the presented problem and its results significantly depend on:
- surface condition
- wettability
- surface tension
- viscosity of the liquid
and in a numerical sense - from the quality of the mesh. These aspects are not described.
Besides, I have some remarks:
- The adoption of the fluid model for blood as Newtonian fluid does not reflect the specific behavior of the blood in the vessel
- The data in Table 2 in my opinion do not come from reference 66. The selection criteria should be commented on.
- Figures 5-16 have an unreadable legend. Invisible units. It would be good to adopt the same range everywhere. The differences of the flow would be apeared.
- In line 200 "haemodynamic" may be more popular "hemodynamic"
- In equations (3 and 4) the operator in the form "." needs some explanation.
Author Response
We would like to thank the reviewer for his comments and suggestions for the authors. We found them to be very helpful and insightful regarding the presented work. Based on the comments and suggestions we have revised the manuscript for typos in the manuscripts and any grammatical errors. The responses to the comments and suggestions are listed below and have been highlighted in blue in the manuscript:
- Comment: the presented problem and its results significantly depend on: surface condition, wettability, surface tension, viscosity of the liquid.
- Comment: The adoption of the fluid model for blood as Newtonian fluid does not reflect the specific behavior of the blood in the vessel.
Response: the surface conditions, wettability, surface tension, and viscosity of the liquid are now highlighted in the manuscript lines [85-90]. We list assumption that are related to the simulation noting the use of Newtonian fluid, effect of surface roughness and why specific assumptions were made. The rationale for the assumptions is based on previously experimental results that were published, and to present a simple and inexpensive strategy for simulation.
- Comment: in a numerical sense - from the quality of the mesh. These aspects are not described.
Response: the quality of the mesh was tested during the simulations as highlighted in the lines [163-167] through using a patch-conforming method and ensuring an accurate mesh independence study.
- Comment: the data in Table 2 in my opinion do not come from reference 66. The selection criteria should be commented on.
Response: the references have been corrected in Table 2 as noted, and the selection of parameters were based on the references and basing of values from hemoglobin in terms of the molecular weight.
- Comment: figures 5-16 have an unreadable legend. Invisible units. It would be good to adopt the same range everywhere. The differences of the flow would be appeared.
Response: Figures 5-16 have been reconstructed to improve the legend quality. The range has been used to understand the higher and lower boundaries of the data and connect it the blood flow values boundary conditions.
- Comment: in line 200 "haemodynamic" may be more popular "hemodynamic"
Response: comment in line [200] was corrected and “fluid metrics” were used instead.
- Comment: In equations (3 and 4) the operator in the form "." needs some explanation.
Response: the operator used “.” has been used as covered as literature and it is the multiplication operator.

Reviewer 2 Report
This study uses computational fluid dynamics simulation to investigate the effects of blood flow velocity and structural parameters (e.g., pore size and distribution) of scaffolds on some parameters relevant to the performance of scaffolds such as pressure drop, permeability and wall shear stress on two kinds of scaffolds with different structures. This manuscript is presented clearly. However, more details must be provided and some points must be clarified before the possible publication. The reviewer sincerely hopes the following comments and suggestions can help the authors to further improve the manuscript. Thanks for this work, and best wishes.
1. The Introduction section presents a nice introduction to the background of this topic. However, the authors didn’t clearly describe what kinds of research questions, problems or unmet needs that encourage them to conduct this study. In other words, the motivation is not clear. Please improve.
2. The description of the purpose of the study (the last paragraph in the Introduction section) should be presented in more detail.
3. Lines 84 and 87: The reviewer thinks “Figure 2” in line 84 and “Figure 3” in line 87 should be “Figure 1” and “Figure 2”, respectively.
4. There are many parameters with the designated values in this computational simulation, and there is an essential need to provide the rationale to design the values of those parameters. For example:
(1) For the first configuration: Why the pore sizes are designed as 300, 350 and 450 µm? Why the number of filament layers, the filament diameter and the layer thickness are designed as 12, 350 µm and 330 µm?
(2) For the second configuration: Please describe the reason for designing 6 for the number of rings, and the reason for designing the pore size for each ring of the graded pore structure.
5. Line 92: what is the difference between the volume of the scaffold and the volume of the solid part? Is the volume of the scaffold defined as the volume within the most outer boundary of the scaffold that include the void within? Please define more clearly.
6. Line 97, regarding the term “unit volume”: Does this volume mean the volume of the scaffold or the volume of the solid part?
7. Line 112: Please describe the reason for assuming the blood is Newtonian in this study. The blood should be regarded as the non-Newtonian fluid.
8. Lines 157-161: Please provide more details regarding the content in this paragraph. Specifically, what is k-w model, and what is the relationship between this model and the prediction of cell growth and viability in the scaffold?
9. Generally, the writing is fine, but a careful revision is still needed for further improving the readability.
Author Response
We would like to thank the reviewer for his comments and suggestions for the authors. We found them to be very helpful and insightful regarding the presented work. Based on the comments and suggestions we have revised the manuscript for typos in the manuscripts and any grammatical errors. The responses to the comments and suggestions are listed below and have been highlighted in blue in the manuscript:
- Comment: The Introduction section presents a nice introduction to the background of this topic. However, the authors didn’t clearly describe what kinds of research questions, problems or unmet needs that encourage them to conduct this study. In other words, the motivation is not clear. Please improve.
Response: a paragraph has been extended to explain the rationale of the work, and the challenge in doing experimental work for blood flow through 3D printed scaffolds line [64-67].
- Comment: The description of the purpose of the study (the last paragraph in the Introduction section) should be presented in more detail.
- Please describe the reason for assuming the blood is Newtonian in this study. The blood should be regarded as the non-Newtonian fluid
Response: the final paragraph has been rewritten, and assumptions of the work have been added to add more detail to the purpose and what strategy has been adopted in the work. The rationale for the assumptions is based on previously experimental results that were published, and to present a simple and inexpensive strategy for simulation. The differences in assumptions have also been noted on how can they affect the results, Line [80-90].
- Comment: Lines 84 and 87: The reviewer thinks “Figure 2” in line 84 and “Figure 3” in line 87 should be “Figure 1” and “Figure 2”, respectively
Response: the figure caption has been corrected.
- Comment: For the first configuration: Why the pore sizes are designed as 300, 350 and 450 µm? Why the number of filament layers, the filament diameter and the layer thickness are designed as 12, 350 µm and 330 µm?
- Comment: Please describe the reason for designing 6 for the number of rings, and the reason for designing the pore size for each ring of the graded pore structure.
Response: The designs specifications including pore sizes, filament layers and diameters for all cases have been based on previously published work. Here we use the same values to compare numerical and experimental results.
- Comment: what is the difference between the volume of the scaffold and the volume of the solid part? Is the volume of the scaffold defined as the volume within the most outer boundary of the scaffold that include the void within? Please define more clearly.
- Comment: regarding the term “unit volume”: Does this volume mean the volume of the scaffold or the volume of the solid part?
Response: We have amended in line [104], and the volume of the scaffold is the intended meaning, and it is the solid volume of material used to print, excluding the porosity.
- Comment: Please provide more details regarding the content in this paragraph. Specifically, what is k-w model, and what is the relationship between this model and the prediction of cell growth and viability in the scaffold?
Response: Line [168-172] we have added a line to correct the sentence where the k-w model is used to account for the turbulence and to accurately represent the behaviour of blood.

Reviewer 3 Report
The goal of this study is to assess the simulation approach to investigate and visualize different blood flow velocities and their corresponding influence on the haemodynamic performance of 3D printed bone scaffolds with different configurations and pore sizes.
Authors also showed that the anatomically designed scaffolds allow the best fluid flow condition suggesting also improved biological performance.
The study is original and allows an insight into a relevant clinical topic. The methods applied are adequate and the knowledge produced is relevant to the field of Bone Tissue Engineering scaffolds. This research is under the scope of this Journal.
However, there are some aspects which need to be improved in the various sections of the manuscript.
- Correct typos in all manuscripts.
(Abstract)
- Identified the aim of the study in the abstract.
- In the results, is important to show more information. Please add some of the p-values.
- Add also conclusions.
(Keywords)
- Please more keywords, and order these keywords / Mesh Terms alphabetically for a standardized presentation of the keywords.
(Introduction)
- What is the importance of this review study? Which results are comparable with other articles? What has this study been new?
- Regeneration bone defects with scaffolds of the pores or the space provision versus compacts materials, please read this article, Palma et al. (2010, New formulations for space provision and bone regeneration. Biodental Eng. I, 71-76) reported the influence of different formulations of bone grafts in providing an adequate scaffold, thus emphasizing the importance of the three-dimensional distribution of particles and also space provision for new bone formation. The scaffolds are crucial for bone regeneration in critical-size defects. For this reason, the authors need to support the necessity of using Scaffolds in “Critical size defects” by example the animal studies. Please read these references. https://doi.org/10.3390/molecules26051339, https://doi.org/10.1111/j.1600-0501.2011.02179.x.
(Results)
- Improve the resolution quality of all figures and graphs (and a presentation). The font/language in the figure/caption is different from the text. Please, standardize the size and the font in the figures with the font of the manuscript.
(Discussion)
- Please, identified what was the strength(s) and limitations of this study? And also, implications for future perspectives before the conclusions.
(Conclusions)
- The conclusion section should more thoroughly summarize the results, this section is too long and sometimes too ambiguous.
(References)
- Check the reference MDPI format in the manuscript and the references. The references have a different format one the manuscript presentation.
Author Response
We would like to thank the reviewer for his comments and suggestions for the authors. We found them to be very helpful and insightful regarding the presented work. Based on the comments and suggestions we have revised the manuscript for typos in the manuscripts and any grammatical errors. The responses to the comments and suggestions are listed below and have been highlighted in blue in the manuscript:
- Comment: Identified the aim of the study in the abstract.
Response: The abstract has been amended and a clearer aim has been added, line [17-19].
- Comment: In the results, is important to show more information. Please add some of the p-values. Add also conclusions.
Response: Figure 4 has been amended to add the p-values highlighting when there is an increase in blood flow velocity, there are smaller differences between the permeability of the scaffolds. Line [203-208]
Please more keywords, and order these keywords / Mesh Terms alphabetically for a standardized presentation of the keywords.
Response: more keywords have been added and they were ordered in alphabetical order as requested for standardized presentation.
- Comment: What is the importance of this review study? Which results are comparable with other articles? What has this study been new?
- Comment: Please, identified what was the strength(s) and limitations of this study? And also, implications for future perspectives before the conclusions.
Response: Two paragraphs have been added, line [314-327]. This highlights the importance of the study, the strategy used, and how that effect the results. The study has been compared to other articles both in methodology and results to understand the significance of the work. Second paragraph highlights the significance of this work based on previous published experimental results with future perspectives to improve the work.
- Comment: Improve the resolution quality of all figures and graphs (and a presentation). The font/language in the figure/caption is different from the text. Please, standardize the size and the font in the figures with the font of the manuscript.
Response: Figures 5-16 have been reconstructed to improve the legend quality. The size and font have been also corrected to that of the manuscript.
- Comment: The conclusion section should more thoroughly summarize the results, this section is too long and sometimes too ambiguous.
Response: The conclusion has been reworded to highlight the results and to reduce its length, with suggestions for future work based on the presented results. Line [331-343]
- Comment: Check the reference MDPI format in the manuscript and the references. The references have a different format one the manuscript presentation.
Response: We have changed the style of the references.

Round 2
Reviewer 1 Report
Thank you for the answers. I accept this paper in this form.
Author Response
We would like to thank the reviewer for his insights and helpful suggestions which have improved the quality of the work and enhanced the presentation of the results.
Reviewer 2 Report
The authors must add every response to the revised manuscript and cite the relevant references. In addition, a professional editing of English writing style is needed.
Author Response
We would like to apologize for any misunderstanding in the previous report and extend our thanks again to the reviewer for his insights and suggestions. The comments were grouped and responded to accordingly as we believed some of the points were similar and one response can be addressed together. Changes to the previous comments were highlighted in blue. Based on the comments and suggestions we have revised the manuscript again to improve the writing and confirm the use of the correct references.
Reviewer 3 Report
The authors improve the manuscript, following the reviewer's indications. Congratulations!
Note: The reference number 30 is incomplete, please add the complete reference below.
(30. Palma, P. J., Matos, S., Ramos, J., Guerra, F., Figueiredo, M. H., & Krauser, J. (2010). New formulations for space provision and bone regeneration. Biodental Eng. I, 1, 71-76. WOS:000282776500012; SBN 978-0-415-57394-8.)
Author Response
We would like to thank the reviewer for his insightful comments and helpful remarks which improved the quality of the manuscript and the presentation of the results.
Comment: The reference number 30 is incomplete, please add the complete reference below.
Response: Reference [30] has been amended to the correct format as indicated by the reviewer.